# Heterogeneity in Circulating Tumor Cells: The Relevance of the Stem-Cell Subset

**DOI:** 10.3390/cancers11040483

**Published:** 2019-04-05

**Authors:** Chiara Agnoletto, Fabio Corrà, Linda Minotti, Federica Baldassari, Francesca Crudele, William Joseph James Cook, Gianpiero Di Leva, Adamo Pio d’Adamo, Paolo Gasparini, Stefano Volinia

**Affiliations:** 1Department of Morphology, Surgery and Experimental Medicine, University of Ferrara, 44121 Ferrara, Italy; gnlchr@unife.it (C.A.); crrfba@unife.it (F.C.); mntlnd@unife.it (L.M.); bldfrc1@unife.it (F.B.); crdfnc@unife.it (F.C.); 2School of Environment and Life Sciences, University of Salford, Salford M5 4WT, UK; w.j.j.cook@edu.salford.ac.uk (W.J.J.C.); G.DiLeva@salford.ac.uk (G.D.L.); 3Department of Medicine, Surgery and Health Sciences, University of Trieste, 34127 Trieste, Italy; adamopio.dadamo@burlo.trieste.it (A.P.d.); paolo.gasparini@burlo.trieste.it (P.G.); 4Institute for Maternal and Child Health—IRCCS “Burlo Garofolo”, 34137 Trieste, Italy

**Keywords:** CTC, EMT, stemness, CSC

## Abstract

The release of circulating tumor cells (CTCs) into vasculature is an early event in the metastatic process. The analysis of CTCs in patients has recently received widespread attention because of its clinical implications, particularly for precision medicine. Accumulated evidence documents a large heterogeneity in CTCs across patients. Currently, the most accepted view is that tumor cells with an intermediate phenotype between epithelial and mesenchymal have the highest plasticity. Indeed, the existence of a meta-stable or partial epithelial–mesenchymal transition (EMT) cell state, with both epithelial and mesenchymal features, can be easily reconciled with the concept of a highly plastic stem-like state. A close connection between EMT and cancer stem cells (CSC) traits, with enhanced metastatic competence and drug resistance, has also been described. Accordingly, a subset of CTCs consisting of CSC, present a stemness profile, are able to survive chemotherapy, and generate metastases after xenotransplantation in immunodeficient mice. In the present review, we discuss the current evidence connecting CTCs, EMT, and stemness. An improved understanding of the CTC/EMT/CSC connections may uncover novel therapeutic targets, irrespective of the tumor type, since most cancers seem to harbor a pool of CSCs, and disclose important mechanisms underlying tumorigenicity.

## 1. Introduction

Despite recent medical advances, metastasis, tumor relapse, a lack of effective therapies, and drug resistance remain the major causes of death for tumor patients [1,2,3,4,5]. Research on circulating tumor cells (CTCs), which have been detected in the majority of epithelial cancers [6,7], is a dynamic field of translational and basic research, with more than 270 clinical trials having evaluated CTC enumeration as a biomarker [8]. Further, several studies on CTCs aimed to understand critical pathways that mediate cancer dissemination, and may not be apparent in primary or metastatic tumors.

Cancer metastasis occurs when tumor cells dissociate from a primary tumor and migrate to distant organs through the peripheral vasculature [1,9,10,11,12]. Cancer cells within the primary tumor lose cell-cell adhesion and acquire features of invasiveness and motility due to epithelial-to-mesenchymal transition (EMT), leading to their intravasation and survival in blood or lymph vessels, and finally their extravasation at distant sites; here they are subjected to mesenchymal-to-epithelial transition (MET), and proliferate to establish metastatic lesions [12,13,14]. Whether the release of CTCs into the circulation is a predetermined biological process still remains a debatable matter. Cancer cells can enter circulation long before a tumor is diagnosed; the majority of cells die and only a minor fraction contains viable metastatic precursors that infiltrate organs and survive as disseminated seeds for eventual relapse [1,15]. Therefore, metastatic colonization might be a highly inefficient process, but once it is established, current treatments generally fail to provide persistent responses [12].

Tumors are thought to arise from mutant normal stem cells in their native niches, or from the progeny of cells that retain their tumor-initiating capacity [16,17]. Once at distant sites, these cells survive and proliferate through interactions with specialized niches [18]. Data from a number of studies demonstrate that CTCs with stemness properties exist and represent the most aggressive tumor cells in circulation [19,20].

In this review, we will discuss current concepts and will address questions relative to the heterogeneity of CTCs and the presence of cancer stem cells circulating in the bloodstream (See Table 1).

## 2. Heterogeneity of CTCs

CTCs are very rare cancer cells released from a primary tumor. A lot of evidence documents that CTCs represent a heterogeneous pool of tumor cells [8], with highly variable, but generally short, survival times; nevertheless, the pool of these cells in the bloodstream can be continuously replenished by the release of replicating tumor cells in metastatic foci, and thus relates to cancer dormancy [21,22]. To date, relatively little is known regarding the number of CTCs in cancers, their molecular and biological heterogeneity, and their functionalities [9]. Although heterogeneity in CTCs has not yet been fully defined, a fraction of these cells are thought to be viable metastatic precursors capable of initiating a clonal metastatic lesion [9]. A subset of CTCs is represented by inactive non-cycling cells, termed dormant cancer cells [23], which act as latent tumor-initiating seeds and eventually reawaken, and might not respond to chemotherapeutics used in clinics [8,12]. By uncovering phenotypes of CTCs, CTC heterogeneity would be dissected in relation to metastatic competence [19,24].

Recently, clusters of CTCs, consisting of 2–50 cells, have been detected in the vasculature of patients with cancers of different origins, which arise from oligoclonal tumor cell groupings, including breast, prostate, pancreatic, glioblastoma, head and neck cancers [24,25,26,27,28,29,30,31,32,33]. Cells within micro-clusters might be protected either from anoikis, associated with the loss of cell-cell adhesion and attachment to the basement membrane, or shear stresses in the circulation [25,26,27]. Both the structural deformability of aggregated cells and the presence of vascular shunts may allow CTC-clusters to circulate, while inducing a rapid clearance within distal tissues [25]. Although rare, CTC-clusters present a dramatically enhanced metastatic potential, as confirmed in mouse models, and by the adverse prognosis in patients with high numbers of CTC-clusters [12,30,34]. Additionally, neutrophils directly interact with CTCs, defining CTC–neutrophil clusters, and support cell cycle progression in circulation. This interaction in turn leads to more efficient metastasis generation and represents key vulnerabilities of the metastatic process for drug targeting in breast cancer [35]. Mesenchymal transcripts were found to be expressed in tumor cells from CTC-clusters in a human breast tumor, conferring both migratory and stem-like properties [32]. A recent study has suggested that the coexpression of EpCAM, CD44, CD47, and MET in CTCs identifies a subset of cells with increased metastatic capacity [36]. Novel mediators of metastasis have been identified by comparing clusters of tumor cells, captured by using a negCTC-iChip, with single CTCs from breast cancer patients: single cell-resolution RNA sequencing demonstrated the significantly divergent expression of only a few genes, including plakoglobin [25]. Plakoglobin functions as an intercellular tether that confers added metastatic potential to tumor cells in the circulation. Knockdown of plakoglobin inhibits CTC-cluster generation and lung metastases, while both abundance of CTC-clusters and high plakoglobin levels denote adverse outcome in breast cancer patients [25], pointing to CTC-clusters as critical mediators of cancer metastasis.

### 2.1. Partial Mesenchymal Transition in CTCs

Recent data support the hypothesis that CTC profile changes occur during tumor cell dissemination, in order to enter into the vasculature, survive in circulation, extravasate, and generate secondary tumors [19,37,38,39]. Thus, in response to signaling proteins released by stromal cells, tumor cells undergo reversible phenotypic change, described as epithelial-to-mesenchymal transition (EMT) [12,39,40,41,42,43]. EMT is also required in tissue repair; consistently, the metastatic process is at least in part similar to tissue regeneration [14,40,44]. EMT results in a reduced expression of epithelial markers, and an increased plasticity and capacity for migration and invasion, as well as a resistance to anoikis and apoptosis, and senescence, which are hallmarks required for CTC survival and dissemination [14,29].

Studies of the effects of EMT in CTCs have suggested that mesenchymal transformation may facilitate the initial steps of the metastatic cascade, but may reduce their competence to initiate overt metastases [19,45,46]. Disseminated cancer cells may at least partially revert to an epithelial phenotype to promote adhesion and proliferation in distal sites, through mesenchymal–epithelial transition [2,44,47,48]. Cancer-associated EMT may therefore not represent a complete interconversion of phenotypes, or an irreversible commitment to a mesenchymal state. Indeed, the existence of a metastable or partial EMT phenotype, with both epithelial and mesenchymal features, adapts to the concept of a highly plastic stem-like state, reversible and responsive to the microenvironment [19].

Several groups investigated this phenotype in breast cancer [8,37,49]. At least one of a selected pool of EMT markers (Twist 1, Akt2, or PI3Kalpha) were observed both in CTC-positive and CTC-negative cancer patients. Overexpression of EMT-initiating transcription factors (TF), including Twist, Snail1, Slug, Zeb1, and FoxC2, was detected in CTCs, even if in a minority of patients [39]. The EMT markers fibronectin and/or vimentin were expressed more frequently in cytokeratins (CK)-negative than in CK-positive samples [50,51]. Of note, patients receiving neoadjuvant therapy were more likely to present a higher expression of EMT markers, predicting worse prognosis more accurately than epithelial markers in patients with CK-negative CTCs [39]; thus, CTCs, subjected to EMT, might be invasive cells resistant to neo-adjuvant therapy.

### 2.2. Epithelial–Mesenchymal Transition and Stemness

Several lines of evidence have coherently confirmed the notion that the metastatic potential of a tumor is due to a low number of a minor subpopulation of cancer cells (~1%)—termed cancer stem cells (CSCs)—in primary tumor tissue, that exhibit two defining abilities, to self-renew and to efficiently regenerate the phenotypic heterogeneity of a parental tumor, and which are responsible for initiating overt metastases [14,49,52,53,54,55,56,57,58]. Studies of neoplastic tissues have provided evidence of stem-like cells within tumors [53,54,59,60], of breast (CD44+CD24−/low) [61,62], pancreatic (CD44+CD24+ESA+) [63], brain (CD133+) [60], head and neck (CD44+) [64], and colon (CD133+ and CD44+ESAhigh) [59] cancer. CSCs have been implicated in both initiating primary tumors and in the seeding and establishment of metastases [53,65,66]. Recent studies have pointed to a crucial link between EMT and the acquisition of stem cell properties [14,67]. In addition to conferring migratory and invasive potential, the induction of EMT in immortalized and transformed human mammary epithelial cells significantly enhanced their self-renewal, tumor-initiating capabilities, and led to the expression of stem-cell markers, associated with breast CSCs [14], suggesting that passage through EMT is an alternative and/or additional step in tumorigenesis. Indeed, the induction of EMT has been associated with expression of the CD44+CD24−/low antigenic profile, which, in breast tumors, defines a subgroup of enriched cancer cells with stem-like properties, as well as mesenchymal traits and cancer stem cell properties, including self-renewal capabilities and resistance to therapies [14,53]. These cells can self-renew in vitro to generate both CD44highCD24low cells and CD44lowCD24high cells, providing further evidence of stemness properties [53]. Aberrations in signaling networks and transcription factors implicated in EMT have been documented in breast cancer progression [40,43,68], and numerous conserved genes in the signature of resident stem cells are considered EMT-associated genes [69]. Consistently, the gene expression profile of CD44+CD24−/low breast cancer cells more closely resembles that of CD44+CD24−/low cells from normal breast tissue [70]. This undifferentiated phenotype of CSCs is evident in basal-like breast tumors [71], which exhibit EMT features accounting for their clinical seriousness [72]. Further, the gene expression signatures of stem cells from normal tissues and of the highly aggressive metaplastic and claudin-low histological breast tumor subtypes share strong similarities with gene expression profiles of cells upon the induction of EMT and cancer stem cells [14,42,69,73].

Both in vitro and in vivo experiments have suggested that long-term maintenance of the EMT/stem cell state depends on continuous EMT-inducing signals [14]. EMT is orchestrated by integrated networks of signal pathways and EMT-related TFs, including TGFβ, NOTCH, and Wnt [40,43,74]. Exposure of tumor cells to both cytokines TGFβ and TNFα induced EMT and generated cells with a stable breast cancer stem cell phenotype, enriched in CD24−low/CD44+ cells, with a shift to the claudin low molecular subtype [14,40,74]. Targeting proteins required for induction of the claudin-low subtype could be a therapeutic strategy for improving the survival of breast tumor patients, regardless of the primary subtype. TGFβ represents a paradigm of duality in cancer [75]. Of note, the initial tumor suppression effect due to TGFβ is coupled to EMT; however, although the bulk of tumor cells die, EMT-permissive cells can survive from the TGFβ-imposed bottleneck [75]. Moreover, tumorigenicity linked to EMT is a trait that must be selected during carcinoma progression. In accordance, EMT is not always connected with effective metastatic dissemination [74,75,76].

Several TFs and miRNAs have been implicated in linking EMT to the acquisition of stem cell properties. EMT-inducing TFs have been confirmed to confer malignant traits in neoplastic cells [14,77], and the acquisition of stem cell properties of self-renewing and tumor-initiating, thus generating highly aggressive tumors with an EMT phenotype and stem cell features, and providing evidence for the emergence of cells with combined EMT/CSC phenotypes [14,43,67]. The discovery of miRNAs has added an additional level of complexity in EMT, metastasis, and stemness [78,79]. Downmodulation of the miR-200 family has been extensively documented to occur during EMT [80], and has been described both in invasive pancreatic tumors, inducing the generation of migrating CSCs [78], and in normal and breast cancer stem cells, reinforcing the links between normal stem cells and CSCs [78,79,80]. Additionally, expression levels of let-7, which is poorly expressed in mesenchymal tumors and targets HMGA2, encoding a DNA-binding protein implicated in mesenchymal cell differentiation, are reduced in breast CSCs, but increased during differentiation [81].

### 2.3. Putative Stem Cell-Like Phenotype in CTCs

EMT enables the detachment of tumor cells from a primary site, but it can also be induced after CTCs enter the vasculature [82]. Only recently, different subsets of CTCs have been identified as mesenchymal [37,83]. However, exclusively mesenchymal CTCs seem to be unable to establish metastases [19,46]. The current hypothesis is that tumor cells with an intermediate phenotype between epithelial and mesenchymal present the highest plasticity to adapt to secondary sites, and constitute cancer stem cells [84]. Consistently, CTCs expressing EPCAM, CD44, CD47, and MET, once injected in immunodeficient mice, generated metastases with an EPCAMlow/METhigh/CD47high/CD44high phenotype, thus containing functional metastasis-initiating cells [36]. Therefore, the detection of such CTCs is of the utmost importance [85].

The use of EpCAM in current CTC detection methods, thus effectively pre-selecting for epithelial CTCs, is a matter of debate, and high expression has been confirmed in CSC from breast, colorectal, and pancreatic tumors [57]. The metastatic potential of EpCAM-positive CTCs has been extensively confirmed in xenograft models [36,86,87], providing the proof of principle that CTCs captured by the CellSearch system are capable of initiating metastasis, at least in murine models. On the contrary, the CellSearch method did not recognize aggressive breast cancer cells [8], and it suffered from a relatively low sensitivity: only a fraction of patients with metastatic cancer scored positive for any CTCs, with a median yield of approximately 1 CTC per milliliter and typically low purity [88].

Although the role of EMT in CTCs is still controversial, EMT signatures have been observed in at least a subset of tumor cells with stemness properties [12,20,39,89,90,91]. However, studies in cancer models suggest that EMT could be dispensable for establishing metastasis, despite the fact that it contributes to the aggressiveness by increasing chemoresistance [12,76,92]. Some investigators reported that CSCs markers were enriched in CTCs with mesenchymal features and had prognostic relevance, in breast tumors at different stages (stage I–III) [93,94]. These CSC phenotypes need to be further evaluated, also in patients with early stage cancer, and prospectively related to the occurrence of overt metastases. Furthermore, CSCs might develop after the initial diagnosis of a tumor, so serial postoperative blood monitoring might be required to identify CSCs [19].

A selection of the more aggressive cells generates clinical evidence of metastasis [58]. Coherently, a significant genetic heterogeneity in early disseminated tumor cells (DTC) in early breast cancer contrasts with homogeneity in tumor cells from metastases [95]. Subsets of CTCs/DTCs demonstrate a stem cell-like phenotype, which confer stemness and pleiotropic roles in cell adhesion, migration, and homing [96,97]. Only a small subset of CTCs, which migrate into vasculature, termed circulating cancer stem cells (CCSCs), with putative stem cell progenitor phenotypes, present high invasiveness and effectively generate metastasis [29,98,99,100]. Blood-monitoring studies in tumor patients have shown that some CTCs can survive chemotherapy, as postulated for cancer stem cells [19]. CCSCs have been identified in several types of cancer, including metastatic breast cancer [21], lung carcinoma [101], hepatocellular carcinoma [102], colorectal cancer [103], and gastric cancer [104]. Specific biomarkers associated with stemness detected in CTCs include BMI1 for lung CCSCs [101], CD90+CXCR4+ for hepatocellular CCSCs [105], and CD44 variant exon 9 (CD44v9) for colorectal CCSCs [103].

Stem cell and epithelial-mesenchymal transition markers are frequently overexpressed in CTCs of metastatic breast cancer [49]. Recently, in a breast tumor, a stem cell-like population, defined by the presence of CD44 and absence of CD24, has been identified [53], which might disseminate into the circulation and escape therapy [106], and presents an expression profile associated with metastatic relapse [66]. It has been reported that the modulation of HER2 signaling can increase the cancer stem cell pool and might be required for its maintenance; in fact, a strong link between HER2- like-tumors and stem cells has been observed [107]. Preliminary data identified a breast cancer stem cell-like phenotype in CTCs, with increased resistance to chemotherapy and decreased proliferation in circulation [93]. ALDH1, a marker of normal and neoplastic breast stem cells [49,65], has been confirmed to be overexpressed in 70% of CTCs, and associated with therapy failure [49]. At least one or more of the EMT markers Twist, Akt2, and PI3Kalpha, and ALDH1 were detectable in a great proportion of CTCs, thus identifying the highly tumorigenic subset of EMT-associated breast CSCs [49]. This subset of CTCs is indicative of therapy-resistance and inferior prognosis in metastatic tumor patients, with clinical relevance [49]. Of note, ALDH1 and EMT markers were measured even before the detection of CTCs in circulation, as assessed through the positivity in RT-PCR to at least one of the transcripts HER2, MUC1, and EpCAM [49]. Consistently, the stem cell-like phenotypes CD44+CD24−/low and ALDH1highCD24−/low have been identified in 35.2% and 17.7% of CTCs, respectively [58], and the subset of CD44+CD24−/low CTCs among ALDH1-highly positive CTCs present an even higher tumorigenic potential [65]. Finally, although ALDH1-positive cells represented only 5% of cells in tumors expressing ALDH1, ALDH1 staining positivity was associated with a high histological grade and poor clinical outcome [65].

According to the cancer stem cell model, tumor progression and drug resistance should depend on putative cancer stem cells [108], and their presence should correlate with a worse outcome [65], despite data being discordant [61,71,109]. Effective elimination of these cells during medical intervention is difficult, due to their self-renewal potency and resistance to chemotherapeutics. Indeed, patients with ALDH1-positive CTCs more frequently did not respond to chemotherapy [49]. Recently, in 86% of metastatic breast cancer patients, CTCs have been demonstrated to express one or more multidrug resistance-related proteins (MRP), and patients with MRP-positive CTCs had a significantly shorter time to progression [110]. Further studies are required to correlate the presence of CD44+CD24− or ALDH1highCD24−/low CTCs with clinical course and disease progression.

CSCs of different metastatic sites might have different phenotypes. In HCC, stem-like CTCs were considered as seeds of metastasis and recurrence [87]. A significant positive correlation was measured between CD133+ CTCs and serum levels of Annexin A3 (ANXA3), which is highly accurate in diagnosing early-stage HCC, and in monitoring the therapeutic response [111]. Prognosis was worse for ANXA3-positive patients with detectable CCSCs, suggesting that serum ANXA3 could stimulate and maintain the stem cell-like traits of CD133+ CTCs to promote tumor recurrence and metastasis [111]. In breast tumor patients, both CK+CD44+CD24−/low CTCs [58], or CTCs expressing CD44, MET, and CD47, might represent CSCs [36]. Moreover, CTCs from patients with metastatic breast cancer, competent in generating brain metastases, were EPCAM− and expressed HER2, EGFR, NOTCH1, and HPSE [112].

EpCAM-negative CTCs may efficiently avoid organ arrest due to the presence of stem cell and quiescence properties [21]. The molecular switch to differentiate quiescence in malignant CTCs depends on interactions with the tumor microenvironment. Previous studies have established that the loss of the urokinase plasminogen activator receptor (uPAR) and integrin β1 (intβ1) reduces proliferative signals, with a shift from an invasive or metastatic to a dormant tumor state [19,113]. In order to identify CTC subsets with properties related to breast cancer dormancy, CTC subgroups were selected for EpCAM and CD45 negativity and positivity for the CD44+CD24− stem cell signature, and the combinatorial expression of uPAR and int β1 [21]. A high expression of embryonic stem-cell genes was confirmed in the uPAR+intβ1+ CTC subset, compared to uPAR−intβ1− CTCs [21]. Accordingly, in vitro assays confirmed the metastatic competency of uPARintβ1 CTCs, consistent with previous notions [21]. The characterization of the uPARintβ1 CTC subset can be useful to decipher cellular and molecular mechanisms of organ-homing CTCs and cancer dormancy.

## 3. Clinical Relevance of Heterogeneity in CTCs

The liquid biopsy of CTC has received great attention, due to obvious clinical implications for personalized medicine [8,15,19]. Clinical applications of CTC detection have been extensively described in recent reviews [7,15]. Strong evidence for CTCs as prognostic markers of metastatic relapse and progression has been documented [29,139,140,141,142,143,144,145,146]. The clinical utility of CTCs enumeration for treatment decisions is currently being evaluated in interventional studies [147]. CTC detection has been proven to predict prognosis also in early-stage tumor patients, without clinical signs of overt metastases [8,9,15,19,58,148]. Detection of CTC at the time of diagnosis or after surgery for a primary tumor can identify patients at risk of recurrent or metastatic disease [149,150]. Moreover, the patterns of drug susceptibility and responses to therapeutic regimens in individual patients, as tumors acquire new mutations over time, could be noninvasively monitored through the characterization of CTCs, as confirmed in a recent proof-of-concept study of drug sensitivity testing in ER–positive breast cancer CTCs [28]. Most agents administered to prevent relapse are designed to target growing cancer cells rather than quiescent DTCs that predominate in metastatic latency [12]. Research on these cells should help in designing novel therapeutic molecules for targeting and monitoring minimal residual disease, and to assess response to therapy [12].

Unfortunately, the detection of CTCs is hampered by their still uncertain biology. Profiling of CTCs in serial liquid biopsies holds great potential to significantly improve cancer therapy by: (i) determining patient prognosis, (ii) monitoring tumor recurrence and therapeutic responses, (iii) identifying new therapeutic targets in order to prevent metastasis, (iv) elucidating drug resistance mechanisms, and (v) improving current understanding of tumor progression and metastasis.

At present, insufficient data on the clinical relevance of stem-like CTCs is still present in the literature, mainly due to technical hurdles in detecting the very rare stem-like tumor cells among CTCs. Nonetheless, as novel non epithelial profiles of CTCs have started to emerge, in this manuscript we underline their relevance to patient management. It is possible that, as for the mesenchymal CTCs, stem-like CTCs will also be proven to bear clinical implications, as suggested from the reported early studies.

The development of single cell analysis technologies has demonstrated a great heterogeneity in the genetic make-up of CTCs, compared to primary tumor and metastatic cells [19,151,152,153,154]. Exclusive genetic mutations of CTCs have been identified, while additional mutations in CTCs were also detected in both subclones of primary tumor and metastatic tissues, as assessed in diverse tumors [155,156]. Discordances have been reported in the expression of cancer-specific antigens between CTCs and primary tumors [153,157]. CTCs represent an effective alternative to invasive tumor biopsies, thus constituting the actual targets of adjuvant therapies, and point out the emergence of resistant clones and patterns of early disease recurrence and therapy responses [12]. CTCs in women initially diagnosed with advanced ER-positive/HER2-negative breast cancer acquired an HER2-positive subpopulation after multiple courses of therapy for recurrent metastatic tumors [154,158]. Although they have comparable tumor-initiating potential and cell plasticity and expression profiles of ALDH1, these coexisting discrete chemotherapy-sensitive HER2+NOTCH1− and NOTCH inhibitor-sensitive HER2−NOTCH1+ CTCs interconvert, thus a dual treatment may be required for an effective response [153]. Also, markers relevant to the breast cancer brain-metastasis-selected CTC profile have been reported [112], and the uPARintβ1 CTC subset may prospectively identify patients at high risk [21]. In breast cancer, the oncogene PIK3CA is closely associated with the reactivation of multipotency [159], and a complete response is more difficult in HER2-positive tumors mutated in PIK3CA after neoadjuvant anthracycline taxane-based chemotherapy plus anti-HER2 treatment, even with a dual anti-HER2 treatment [160]. A multimarker panel based on single CTC analysis, which defines resistant tumor cells in ovarian cancer by detecting epithelial, EMT, and stemness-associated transcripts, has recently been designed [123], confirming that the number of CTCs is negatively correlated with patient overall survival. In metastatic castration-resistant prostate cancer (CRPC) patients, a set of EMT-related genes (PTPRN2, ALDH1, ESR2, and WNT5A) were upregulated in CTCs [137]; upon targeted therapy two distinct CTC subgroups were induced, which differ in both AR genotype and genomic CNV profiles [161]. Thus, serial analyses of CTCs represent a reliable test for the molecular evolution of tumors and may be relevant upon effective targeted cancer therapy that induces drug resistance.

As for EMT in CTCs, the presence of mesenchymal markers can accurately predict clinical outcomes and a poor prognosis [110,162,163]. In HCC metastasis, Twist and vimentin expression can be applied as diagnostic and prognostic biomarkers; of note, the expression levels of Twist, vimentin, and E-cadherin were closely associated with the positivity rates of CTCs in peripheral blood, which indicated that EMT promotes the dissemination of primary HCC cells [131]. In breast cancer patients, several differentially expressed genes, involved in EMT, were identified in CTC, including UPA, IGFR1, VEGFR1, and CD44, and their expression was correlated with tumor grade and metastasis [89]. Activation of the urokinase plasminogen system, which has been associated with poor prognosis [164], and IGFR1 both contribute to recurrence via the inhibition of anoikis [165,166]. VEGFR1, a positive regulator of angiogenesis, participates in tumor progression through the induction of EMT [167]. Expression of CD44 has been detected in breast primary tumors [168] and significantly increased in CTC-enriched samples, in accordance with a previous report [58]. In pancreatic tumor expression of both epithelial and mesenchymal markers, LGFBP5 and ALDH1 have been detected in CTCs [169].

The clinical relevance of mutation analysis of CTCs in assessing prognosis, stratifying patients, defining precise therapies, and monitoring therapeutic efficacy has been confirmed [170]. In breast cancer, the analysis of single CTC with a diagnostic intention identified preexisting cells resistant to ERBB2-targeted therapies [171]. In colorectal cancer, several mutations that induce resistance to anti-EGFR therapy have been reported; CTC monitoring has revealed that the emergence of KRAS mutant clones is a secondary mechanism of drug resistance [172,173]. The acquired S492R EGFR mutation prevents cetuximab binding, and confers resistance [174]. In non–small-cell lung cancer (NSCLC), the diagnosing of EML4-ALK rearrangement, a potent oncogenic driver [175,176], is a critical issue for identifying patients with advanced tumors that eventually respond to crizotinib [177]. A recent study provides the first proof-of-concept that CTCs can be used for highly sensitive and specific diagnostic testing of ALK rearrangement [126]. A unique pattern of ALK rearrangement has been consistently identified in CTCs, despite inter-tumoral heterogeneity [126]. Additionally, results support a role for ALK in generating mesenchymal CTCs [40,43], suggesting that CTCs with a unique ALK rearrangement and a mesenchymal phenotype may result from the clonal selection of tumor cells that have acquired the potential to metastasize, such as cancer stem cells [126].

As mentioned above, recent advances in the identification of stem-like signatures in primary tumors and CTC may help to improve the accuracy of diagnosis and predict therapeutic responses and overall survival of patients. Molecular profiling of CSCs has revealed their major implications for clinics [136,178,179,180,181]. Consistently, the phenotypic features of CTCs, including CCSCs, have been analyzed to assess tumor stage, risk of cancer progression and metastases, and disease relapse, and their potential in predicting the overall outcome of patients with locally advanced and metastatic breast, prostate, ovarian, pancreatic, lung, and colorectal cancers has been confirmed [116,120,121,134,162,182,183]. Several biomarkers of stemness have been assayed in CTCs (CD133, CD44, ALDH, and ABC multidrug transporters) [116,120,182,183]. In breast cancer, the expression of Nanog, Oct3/4, Sox-2, nestin, and CD34 on CTCs is directly related to disease progression [118]. Recently, a very small cohort of metastatic breast cancer patients was evaluated after palliative chemo-, antibody—or hormonal therapy; most patients received anthracyclines and taxanes in an adjuvant or metastatic setting before the study’s start, and all patients with HER2+ tumors but one received trastuzumab in the metastatic setting. In this cohort, the expression in CTCs of at least one EMT marker and ALDH1 was detected in 62% and 44%, respectively, of patient non responders to therapy, suggesting their potential use as predictive biomarkers [49]. The hypothesis of plasticity and stemness properties in CSCs has been addressed in endometrial carcinomas (EC) [184], in association with a micro-RNA signature of EMT of down-modulation of the miR-200 family [185]. The presence of endometrial CTC, profiled for stem-cell markers ALDH and CD44, was related to recurrent disease [122]. CTCs detected in patients with advanced lung cancer include cells co-expressing cytokeratin and CD133 [127]. In a cohort of patients with primary and metastatic melanoma, the melanoma antigen melan-A and the stem cell–like marker ABCB5 were expressed in CTCs in 45% and 49% of patients with recurrence after treatment initiation, respectively [138].

A major clinical relevance of CTC early detection is the identification of high-risk subjects that need additional systemic therapies after primary tumor surgery, to prevent metastasis. Currently, the selection of patients is based on the statistical risk of recurrences. Evidence for the presence of CSC in high-risk EC has been reported, and a molecular CTC-phenotype has been identified, associated with plasticity and stemness features [122]. Previously, a six-gene panel has been assessed in CTCs in EC [114], with a potential in terms of diagnosis/prognosis and follow-up, and identification of new therapeutic strategies to limit metastatic dissemination [186]. Several genes related to plasticity were significantly expressed in CTC (ETV5, NOTCH1, Snai1, TGFB1, Zeb1, Zeb2) [122]. Of note, ETV5 up-modulation promoted metastasis in in vivo models and recapitulated the plasticity phenotype of high-risk patients [187]. The sequential assessment of CTC levels during treatment could provide information at early stages about therapeutic efficacy.

Moreover, CSCs, generated through EMT, present intrinsic resistance to conventional chemotherapies [41,188,189,190,191]. Current chemotherapies target the rapidly proliferating cells of the tumor bulk, with expansion of the CSC pool and/or selection of resistant and highly metastatic competent CSCs. Accordingly, in breast tumors, residual cancer cells exhibited expression profiles of cells with combined tumor-initiating and mesenchymal/claudin-low features [190]. Gene signatures of the EMT/CSC phenotype may disclose novel molecular targets for the therapeutic resistance of breast cancers [43,49,52]. In a seminal study, salinomycin, a potassium ionophore, has been identified with selective toxicity against CSCs generated by EMT, reducing the CD44highCD24−/low fraction and the ability of pretreated CSCs to initiate tumors and lung metastases in immunocompromised mice [192]. Further, metformin selectively killed breast CSCs in vitro and in vivo, effectively reducing tumors and preventing relapse in mouse models [193]. Overall, cells with the EMT/CSC phenotype may uncover new therapeutic targets for tumors, irrespective of their subtype.

## 4. Advancement in Methodologies for the Inclusive Detection of Stem-Like CTCs

The rarity of CTCs in blood specimens (as few as one cell per 10^9^ haematologic cells) requires highly specialized technologies for detection and capture, but their low sensitivity has limited the clinical utility of CTC-based diagnostics. A lot of reviews have recapitulated and pointed out a conceptual framework of CTC assays, discussing limits of technologies, in light of the recent data (see Table 2 for a list of the most relevant technologies for CTC detection and characterization) [6,7,8,19,88,194,195,196,197]. Epithelial markers, expressed on epithelial tumors but absent on leukocytes, have been frequently used [198,199,200]. While clinical studies using the CellSearch device have confirmed the prognostic potential of CTC enumeration [201], only a fraction of patients with metastatic cancer score positive for CTCs [7,88,202]. Although first defined as nucleated cells expressing EpCAM, and/or cytokeratin, and lacking the leukocyte antigen CD45, new data suggest even greater heterogeneity in CTCs [203,204,205]. Considering the phenotypic heterogeneity and plasticity of CTCs, CTCs with no/low levels of EpCAM and a more mesenchymal phenotype, such as CSCs, might be missed by EpCAM-based capturing techniques [43,77,88,206,207,208].

To effectively capture all diverse CTCs, additional alternative phenotypes need to be detected, and an automated and high-precision device integrating CTC capturing and sequencing is needed [88,195,209,210,211,212,213,214,215,216]. Due to recent advances in the field of microfluidic bioengineering, novel methods have been introduced to improve the accuracy and sensitivity of CTC isolation, with efficient purification of viable CTCs from unprocessed whole blood, in both epithelial and non-epithelial cancers [26,37,195,197,210,211,212,217]. Once CTCs have been sorted in solution, clinically standardized analyses can be applied, as well as single-cell RNA profiling. To avoid the loss of CTCs, normal haematopoietic cells have been depleted by targeting CD45 [218]. However, not all CD45– cells are tumor cells, and a CK+/CD45+ cell phenotype has been reported in metastatic cancer patients [197,201,215,216]. Examples of technologies used to isolate CTCs through both enrichment and depletion include MACS, a size-dictated immunocapture chip, and the CTC-iChip [88,157,219]. The CTC-chip enables a high yield of capture and purity, although it is technically difficult and not yet standardized for high throughput applications [26,195,196]. The MagSweeper has been used for the genetic profiling of single-cell CTCs, including transcriptional profiling, the detection of mutations, whole exome sequencing, and analyses of stem cells in several cancers [157].

High-throughput microscopy and imaging flow cytometry IFC can be combined with cutting-edge CTC isolation technology. One such approach currently under investigation relies on the concept of an inertial microfluidic device, and staining of CTCs with anti-CD45, anti-cytokeratin, anti-E-cadherin, and anti-vimentin [220]. Once clinically validated, this technology, adapted for EMT and stem cell markers, would provide a technological solution for implementation in clinical diagnostics.

A novel enhanced CTC enrichment strategy was established, which captures heterogeneous EpCAM-depleted CTC subsets by targeting cell surface proteins (CD44, CD47, c-Met, reported to be expressed in CTCs with metastasis-initiating properties [36], Trop2, CD49f, CK8, ADAM8, TEM8, CD146) [36,203,221,222,223,224,225], as well as extracellular matrix components (laminin, collagen I, HA) [206]. In 20 out of 29 EpCAM-depleted fractions from metastatic breast cancer patients, additional EpCAMneg CTCs were identified applying Trop2, CD49f, c-Met, CK8, and/or HA magnetic enrichment. Of note, EpCAMneg cells captured by a CD44 antibody were confirmed to have a malignant nature [206]. Further characterization of identified EpCAMpos/neg and dual-positive subsets will assess CTCs tumor origin. To prove the clinical relevance, a study identifying EpCAMlow/neg CTCs with specific genomic profiles should be designed.

A selective and sensitive detection method for the isolation of both CTCs and CCSCs is required for advancement in precision medicine [29,88,208,214]. Fluorescence-activated cell sorting has been used for CCSC isolation by comparing a CSC marker (CD133 or CD44) with leucocyte CD45 [53,65,208,226,227]. Overall, only a limited success in CCSC detection has been reported by using immune-affinity-based methods with multiple fluorescent probes [214,228]. To address this, a novel nanoparticle-mediated Raman imaging method has been described for the detection and analysis of heterogeneous CTC subtypes, with a high isolation efficiency, which profiles CCSCs in accordance with surface marker expression [214]. The selective detection and accurate analysis of tumorigenic properties of CCSCs and CTCs were demonstrated [214]. Five different RANs, as multifunctional probes designed for the detection, isolation, and further analysis of circulating cells through Raman imaging, were designed that identify different CTC subtypes and CCSC’s stemness, by targeting CD133. Transplanted CCSCs could proliferate (CD133+ and CD44+CD24−/low), as well as differentiate, into subtype-specific breast cancer cells in metastatic cancer modeling, including HER2 positive and basal-B [214]. By detecting CCSC, cancer metastasis and relapse would be predicted, and the heterogeneous tumorigenicity of CTCs interrogated, to facilitate the study of chemotherapy effects.

Stem-cell-like CTCs have also been detected through PCR-based approaches. In CRC patients, the detection of CTCs positive for CEA/CK19/CK20/CD133 expression represented a significant prognostic factor for poorer DFS and OS in Dukes stage C patients, and in stage B patients with unfavorable pathological features [229]. In addition, CEA/CK/CD133+ CTC could be a marker of treatment-resistant stem cells [20]. In a recent study, which aimed to identify stem cell markers for CTSC detection, the differential expression of several stem cell markers was observed between CRC patients’ tumor and matched normal tissue, including OLFM4 and LGR5 [230]. Furthermore, plastin 3 (PLS3) is expressed in EMT-induced CTC in CRC, thus identifying tumor cells with a down-modulated expression of epithelial markers; the detection of such cells was an independent prognostic factor for progression-free (PFS) and overall (OS) survival, also in the Dukes B subset [83]. These findings suggest that OLFM4, LGR5, and PLS3 may be useful for the detection of CTSCs [194].

The presence of CTCs at different stages of prostate cancer has been assessed, using the ProstateCancerDetect kit and the AdnaTest EMT-2/StemCell designed panel [231]. Positive results were mainly seen in patients with metastatic disease; EMT-like gene expression was exclusively detected in metastatic tumors, while ALDH1 was even increased in localized tumors [231]. Thus, in advanced prostate cancer, CTCs mainly express transcripts associated with progression and metastatic disease. Even so, the test’s prognostic and predictive implications remain to be determined. The expression of several markers has been analyzed by immunocytochemistry in CTCs from patients with CRPC. Most of the CTCs co-expressed CD133 with epithelial and mesenchymal markers [116]. A quantitative PCR method has also been used to detect EMT (Twist1 and vimentin) and stem cell genes (ABCG2, CD133, PSCA) in peripheral blood from patients with metastatic prostate cancer; expression of stem cell-related genes indicates poor prognosis, whereas EMT-related expression does not [221]. Recently, we described a method based on a qRT-PCR assay for the identification of novel CTC markers, irrespective of currently used markers and without pre-selection or enrichment, by using patient-derived xenografts (PDXs) models; the genes identified using this method in PDX are now also being tested for the study of CTC heterogeneity in human patients [232].

## 5. Conclusions

In conclusion, the reports we discussed here suggest a compelling need to address the definition of common criteria describing critical subtypes of CTCs. During the progression of cancer, EMT can confer properties of stem cell state and acquisition of malignant traits by more differentiated neoplastic cells. Of clinical relevance, patients with CTCs presenting a stem cell-like or EMT phenotype respond less to chemotherapies and show shorter progression-free survival. Detection of this type of CTC could be a powerful diagnostic tool for patient stratification and for the early determination of failure of a therapeutic intervention. As proof, gene signatures and stem cell–like markers expressed in CTCs have been identified for non-invasive prognosis. The molecular targeting of CSCs and CTCs represents a potential strategy to counteract cancer progression, and to overcome treatment resistance and relapse. Additionally, the identification of CSC and CTC targets might contribute to addressing the chemo-resistance of circulating CSCs, a major problem in cancer biology and therapeutics.

## Figures and Tables

**Table 1 cancers-11-00483-t001:** List of the most relevant mesenchymal and stemness markers for CTCs detection.

Marker	Mesenchymal (M)/Stem-Cell (SC) Marker	Functions and Clinical Relevance	Reference	Year
**Breast Cancer**				
Twist1, Akt2, PI3Kalpha, ALDH1	M (Twist1, Akt2, PI3Kalpha), SC (ALDH1)	Higher expression rates of EMT markers and ALDH1 (62% and 44% of patients, respectively) in metastatic patient non-responders to therapies, compared to responder patients.	[49]	2009
CD44, CD24, ALDH1	SC	Presence, in metastatic patients, of CTCs with stem-like/tumorigenic phenotype CD44+CD24−/low (35.2% of CTCs identified), and a less commonly observed population of ALDH1highCD24−/low (17.7% of CTCs analyzed in seven patients), thus identifying a subset of CTCs with putative stem cell progenitor phenotypes.	[58]	2010
CCNE2, DKFZp762E1312, EMP2, MAL2, PPIC and SLC6A8	M	81% of advanced breast cancer patients with recurrence and 29% of breast cancer patients at initial diagnosis positive for at least one gene.	[114]	2010
ALDH1, vimentin, fibronectin	SC (ALDH1), M (vimentin, fibronectin)	Assessment of molecular profile of CTCs according to expression of ERa, HER2/neu, ALDH1, vimentin, and fibronectin. Detection of ALDH1+ CTCs in 28/61; vimentin+ CTCs in 17/61; and fibronectin+ CTCs in 11/61 of patients. Expression of ALDH1 on CTCs significantly correlated to stage of disease (*p* = 0.01) and to expression of vimentin and fibronectin (*p* = 0.001 for both).	[115]	2011
Twist, vimentin	M	Vimentin- and Twist-expressing CK(+) CTCs identified in 77% and 73%, respectively, of early and in 100% of metastatic breast cancer patients. Higher incidence of CK+vimentin+ and CK+Twist+ cells in patients with metastatic disease compared to early stage breast cancer. Significant correlation (*p* = 0.005) between the number of CTCs expressing Twist and vimentin within the same setting, with a reduction in an adjuvant chemotherapy setting in metastatic tumors.	[38]	2011
Fibronectin, vimentin	M	More accurate prediction of worse prognosis in metastatic patients due to the presence of mesenchymal markers, than expression of cytokeratins alone.	[50]	2011
MRP, ALDH1	SC (ALDH1)	Shorter progression-free survival (PFS) in metastatic patients with a ‘drug resistance’ CTCs profile and expressing MRPs. Statistically significant correlation between the number of MRPs expressed in CTCs and ALDH1. The number of MRPs expressed on ALDH1+ CTC is predictive of poor response to treatment and significantly associated with shorter PFS. ALDH1/MRP-expressing CTCs have a greater tendency to intrinsic drug resistance.	[110]	2011
Vimentin, N-cadherin, O-cadherin, CD133	M (Vimentin, N-cadherin, O-cadherin), SC (CD133)	More than 75% of CTCs in metastatic breast cancer co-express CK, vimentin, and N-cadherin.	[116]	2011
CD44	SC	Clinical relevance of CTC (CD45-EpCAM+ cells) and CTSC (CD45-EpCAM+CD44+CD24− cells). Statistical differences between CTC < 50 and CTC ≥ 50 groups among TNM stages, histology stages, and ER and PR status (*p* < 0.05). Statistical differences between CTSC negative and positive groups within TNM stages and regional lymph node metastasis (RLNM) status (*p* < 0.05).	[117]	2012
Twist, Snail1, Slug, Zeb1, FoxC2	M	Higher expression levels of EMT-inducing TF in patients receiving neoadjuvant therapy with respect to patients who received no neoadjuvant therapy (*p* = 0.003).	[39]	2012
Nanog, Oct3/4, Sox2, Nestin, and CD34	SC	Linear relationship between gene expression of stemness markers and tumor stage (I-IV), as well as specific expression patterns by stage.	[118]	2012
CD44, CD47, MET	SC (CD44), M (CD47, MET)	Functional circulating tumor-initiating cells, with increased metastatic capacity. Correlation of EPCAMlowMEThighCD47highCD44high CTCs number and lower overall survival and increased number of metastases.	[36]	2013
Fibronectin 1(FN1), cadherin 2 (CDH2), serpin peptidase inhibitor, clade E (SERPINE1)	M	Association of mesenchymal CTCs with disease progression: in an index patient, reversible shifts between mesenchymal and epithelial phenotypes accompanied each cycle of response to therapy and tumor progression.	[37]	2013
HER2, EGFR, NOTCH1, HPSE		Identification of a potential signature of brain metastases in CTCs comprising brain metastases selected markers HER2+EGFR+HPSE+NOTCH1+. CTC lines expressing this signature were highly invasive and competent in generating brain and lung metastases when xenografted in nude mice.	[112]	2013
Plakoglobin	M	Intercellular tether that confers added metastatic potential. High levels in breast cancer patients denoting adverse outcomes, while selective knockdown inhibiting lung metastases in mouse model.	[25]	2014
UPAR, intβ1	SC	Identification of DAPI−CD45−EpCAMnegativeCD24−CD44+uPARintβ1 CTC subsets with properties related to dormancy. Embryonic stem-cell gene expression profiling revealed high expression in uPAR+intβ1+ CTC subset, and in vitro assays confirmed the metastatic competency of uPARintβ1 CTCs. uPARintβ1 CTC subset may prospectively identify patients at high risk of brain metastases.	[21]	2015
IGFR1, UPA, VEGFA, VEGFR1	M	Genes expressed exclusively in CTC-enriched samples, identified by profiling a panel of 55 breast cancer-associated genes.	[89]	2016
ALDH1, Twist1	SC (ALDH1), M (Twist1)	Prognostic relevance in metastatic patients of single CSC+/partial-EMT+ CTCs (co-expressing cytokeratin, ALDH1, and nuclear Twist1). Evidence of CSC+/partial-EMT+ CTCs in 27.7% of patients at baseline, and correlation to lung metastases and decreased PFS. Detection of CSC+/partial-EMT+ CTCs as an independent factor predicting for increased risk of relapse. Additional association with reduced OS and increased risk of death in HER-2 negative patients. Significant increase in incidence of CSC+/partial-EMT+ CTCs due to chemotherapy confirmed in HER2-negative patients and in non-responders at the end of treatment.	[119]	2019
**Ovarian, Cervical, Endometrial Cancers**				
CCNE2, DKFZp762E1312, EMP2, MAL2, PPIC and SLC6A8	M	In the cervical, endometrial, and ovarian cancer groups, percentage of positive patients of 44%, 64%, and 19%, respectively.	[114]	2010
Vimentin	M	In ovarian cancers, greater percentage of tumor cells with very low EpCAM expression and high vimentin expression. EpCAM expression significantly lower in the vimentin high group (*p* = 0.0036).	[120]	2010
PPIC	M	Identification of a panel of 11 novel gene markers including PPIC for detection of CTCs with clinical impact in epithelial ovarian cancer (EOC) patients, both before primary therapy and during follow-up. Correlation of CTCs over-expressing PPIC, with an incomplete epithelial phenotype and a more aggressive potential, and resistant to chemotherapy, with an adverse outcome (DFS and OS), independently of clinicopathological parameters.	[121]	2013
ETV5, NOTCH1, Snai1, TGFB1, Zeb1, Zeb2, ALDH1, CD44	M (ETV5, NOTCH1, Snai1, TGFB1, Zeb1, Zeb2), SC (ALDH1, CD44)	Remarkable plasticity phenotype in high-risk endometrial cancer CTCs defined by expression of the EMT markers ETV5, NOTCH1, SnaiI1, TGFB1, Zeb1, and Zeb2, further recapitulated through up-regulation of ETV5 in an EC cell line, and demonstrating an advantage in promoting metastasis in an in vivo mouse model. Expression of ALDH and CD44 in CTCs, pointing to an association with stemness.	[122]	2014
CD44, ALDH1A1, Nanog, Oct4, N-cadherin, Vimentin, Snai2, CD117, CD146	SC (CD44, ALDH1A1, Nanog, Oct4), M (N-cadherin, Vimentin, Snai2, CD117, CD146)	Heterogeneous expression of stem cell- and EMT-associated transcripts in ovarian cancer CTCs. Co-expression of epithelial, mesenchymal, and stem cell transcripts on the same CTC was observed.	[123]	2016
**Lung Cancers**				
Vimentin	M	Greater percentage of tumor cells with very low EpCAM expression and high vimentin expression. EpCAM expression significantly lower in the vimentin high group (*p* = 0.012).	[120]	2010
Vimentin	M	In non-small-cell lung carcinoma (NSCLC), correlation of the presence of CTCs detected by both CellSearch and ISET (vimentin-positive cells with cytological criteria of malignancy) with shorter DFS. Complementary methods for detection of CTCs in preoperative surgery.	[124]	2011
Vimentin	M	Detection of isolated or clusters of dual CTCs strongly co-expressing vimentin and keratin in all metastatic NSCLC patients analyzed, confirming the existence of hybrid CTCs with an epithelial/mesenchymal phenotype.	[125]	2011
Vimentin	M	Expression of vimentin in the majority of cells within circulating tumor microemboli and only in some CTCs; no homogeneous EMT transition in tumor cells within the circulation, both in small cell lung cancer and NSCLC.	[90]	2011
Vimentin, N-cadherin	M	Detection of ALK rearrangement in CTCs of patients with ALK-positive NSCLC, enabling both diagnostic testing and monitoring of crizotinib treatment. CTCs with a unique ALK rearrangement and mesenchymal phenotype may arise from clonal selection of ALK-positive tumor cells which acquired metastatic potential.	[126]	2013
CD133	SC	In advanced metastatic tumors, identification of CD133+ cells, a marker of stem-like behavior in highly tumorigenic cells, in subsets of CTCs isolated using centrifugal forces.	[127]	2013
BMI1, Twist1, CD133, ALDH1A1	M (BMI1, Twist1), SC (CD133, ALDH1A1)	Overexpression in CTCs of ALDH1A1 (10/10 of patients), CD133 (3/10 of patients), BMI1 (7/10 of patients), and Twist1 (3/10 of patients), thus confirming the presence of CSCs in NSCLC, complying with the recent demonstration of tumor initiation capabilities in lung CTCs.	[101]	2016
**Hepatocellular Carcinoma**				
CD90, CD44	SC	Presence of CD45-CD90+ CTCs in 91.6% of patients, generating tumor nodules in immunodeficient mice. The CD90+CD44+ cells demonstrated a more aggressive phenotype than the CD90+CD44- counterpart and generated lung metastatic lesions in mice. CD44 blockade prevented local and metastatic tumor nodules by CD90+ cells.	[128]	2008
ICAM-1	SC	Expressed on a minor cell population in HCC CTCs (0.3%), acting as a marker of HCC stem cells. ICAM-1 inhibitors slow tumor formation and metastasis in mice. Increased numbers of CD45-ICAM-1+ CTCs in patients with HCC correlated with worse clinical outcomes.	[129]	2013
N-cadherin, Vimentin	M	Identification of different individual cell type profiles in CTCs, with distinct clinical implications. The presence of mesenchymal cells correlated to survival, while an increase in epithelial cells was associated with a worse treatment outcome. The shift from mesenchymal to epithelial cell profiles was significantly correlated with shortened TTP in both N-cadherin and vimentin to cytokeratins ratios, respectively.	[130]	2013
Twist, Vimentin, E-cadherin	M	Detection of Twist and vimentin in CTCs from 84.8% and 80.4% of patients, respectively. Significant correlation of co-expression of both in CTCs (in 69.6% of patients) with portal vein tumor thrombus, TNM classification, and tumor size, as promising biomarkers for evaluating metastasis and prognosis. Significant association of E-cadherin, vimentin, and Twist levels in HCC tumors with CTCs.	[131]	2013
CD90, CXCR4	SC	Generation of tumors after serial adoptive transplantations of CD90+CXCR4+ HCC cells into NOD/SCID mice, and more frequent detection of released CTCs.	[105]	2015
CD133	SC	Detectable CD133+ CTC positively correlated with serum ANXA3 level (*r* = 0.601, *p* < 0.001), which is associated with a higher risk of recurrence and shorter overall survival: ANXA3 could stimulate and maintain the stem cell-like traits of CD133+ CTCs to promote tumor recurrence and metastasis.	[111]	2018
**Colorectal Cancer**				
ALDH1, CD44, CD133	SC	Analysis of prognostic value of drug resistance and stemness markers in CTCs from metastatic patients treated with oxaliplatin- and 5-fluoruracil-based regimens. No correlation between the expression of CD44 or CD133 in CTCs and patient outcome; statistically significantly shorter PFS in patients with CTCs positive for ALDH1, survivin, and MRP5, for selection of patients resistant to chemotherapy.	[132]	2010
Plastin3 (PLS3)	M	PLS3-positive CTC independently associated with prognosis in metastatic tumors, particularly in patients with Dukes B and Dukes C CRC.	[83]	2013
CD44 variant exon 9 (CD44v9)	SC	Positive expression of CD44v9 in CTCs in 40% of patients. CD44v9 in CTCs as a factor predicting recurrence, prognosis, and treatment efficacy, both in patients with stage III and stage IV disease.	[103]	2015
CD133, CD54, CD44	SC	Higher survival in patients who underwent resection of the primary tumor and surgical treatment for liver metastasis (*p* < 0.001). Worse survival in patients with high expression of CD133+CD54+ (*p* < 0.001), CD133−CD54+ (*p* = 0.004), and CD133+CD44+CD54+ (*p* = 0.003) CTCs subsets. Prognostic value, especially in survival of CRC patients who did not undergo surgical treatment for metastasis. Carcinoembryonic antigen levels, treatment strategy and CD133+CD44+CD54+ CTCs as independent prognostic factors.	[133]	2017
**Pancreatic Adenocarcinoma**				
Vimentin	M	Vimentin-positive CTCs identified in patients who did not have detectable CTCs by CellSearch, isolated by using the ISET (isolation by size of epithelial tumor cells) filtration device.	[134]	2012
ALDH1A2, IGFBP5	M (IGFBP5), SC (ALDH1A2)	Presence in CTCs of low-proliferative signatures, enrichment of the stem-cell-associated gene ALDH1A2, biphenotypic expression of epithelial and mesenchymal markers, IGFBP5, a gene transcript enriched at the epithelial-stromal interface, and high expression of stromal-derived extracellular matrix proteins.	[135]	2014
**Prostate Cancer**				
ALDH	SC	Elevated clonogenicity and migratory behavior of ALDH-high cells. Competence in generation of distant metastasis and enhanced tumor progression of ALDH-high in preclinical models. Expression of several ALDH isoforms in clinical specimens of primary tumors with matched bone metastases.	[136]	2010
Vimentin	M	Identification of vimentin positive CTCs with BRCA losses, correlated with advanced tumor, invasiveness and recurrence, and shortened DFS.	[51]	2010
Vimentin, N-cadherin, O-cadherin, CD133	M (Vimentin, N-cadherin, O-cadherin), SC (CD133)	In metastatic castration-resistant prostate cancer (CRPC), the majority (>80%) of CTCs co-express epithelial proteins (EpCAM, cytokeratins, E-cadherin), with mesenchymal proteins (vimentin, N-cadherin, O-cadherin), and the stem cell marker CD133.	[116]	2011
IGF1, IGF2, EGFR, FOXP3, TGFB3	M	Commonly observed in CTCs in metastatic tumors, despite heterogeneous expression patterns of individual CTCs. An additional subset of EMT-related genes (PTPRN2, ALDH1, ESR2, and WNT5A) were expressed in CTCs of metastatic CRPC, but less frequently in castration-sensitive cancer.	[137]	2013
**Gastric Cancer**				
CD44	SC	Prognostic significance of CD44-positive CTCs. Patients with higher CD44-positive CTCs were more likely to develop metastasis and recurrence than patients with CD44-negative CTCs. The presence of CD44-positive CTCs and TNM stage were independent predictors of recurrence.	[104]	2014
**Melanoma**				
MLANA, ABCB5, TGFβ2, PAX3d, MCAM	M	Detection of CTCs at all stages and after surgical resection of tumor. Significant prognostic value of expression of ABCB5 and MLANA in inferring disease recurrence. Correlation of MCAM expression with poor patient outcome after therapeutic nonsurgical treatment.	[138]	2013

**Table 2 cancers-11-00483-t002:** List of the current most relevant technologies for CTC detection and characterization for clinical use.

Assay/Technology	Manufacturer	Technology/Process Description	Reference
**Immunoaffinity-Based Methods**			
AdnaTest Select and Detect kits	AdnaGen, Langenhagen, Germany	For enrichment and molecular characterization of CTCs, ensuring high specificity and sensitivity of isolation and detection, by processing multiple samples in parallel. Each kit is designed as a specific immunomagnetic cell-selection system (dependent on the tumor, e.g., MUC1-, EpCAM-Ab-coupled microbeads in breast tumor) for enriching CTC and analysis by RT-PCR of tumor-associated gene expression (positive for at least one of the following markers: MUC1, HER2, EpCAM in breast cancer).	[49,123,233]
CellSearch™	Veridex	Automated immunomagnetic enrichment and staining system for quantification of CTCs in whole blood samples. CTCs are enriched using ferrofluids coupled to anti-EpCAM antibodies, defined as 4′,6-diamidino-2-phenylindole (DAPI)+ cells, and identified by cytokeratin staining using fluorescent anti-CK antibodies, as well as counterstaining with anti-CD45 antibodies. CTC can be enumerated and visualized. Currently, the only diagnostic test cleared by the FDA.	[8,198]
Dynabeads^®^ CD45	Invitrogen, Carlsbad	Superparamagnetic beads covalently coupled to anti-human CD45 antibody for efficient isolation or depletion of CD45+ leucocytes directly from whole blood, buffy coat, or mononuclear cells (MNC) suspensions. It can also be used to enrich epithelial tumor cells.	[234]
CELLection™ Epithelial Enrich Dynabeads^®^	Invitrogen, Carlsbad	Superparamagnetic beads covalently coated with a monoclonal anti-EPCAM antibody. The beads will bind to the tumor cells after a short incubation. The bead-bound cells are separated on a magnet and subsequently released from the beads. Up to 5-log enrichment of human epithelial tumor cells, directly from whole blood, bone marrow, or PBMC, will be obtained, suitable for any downstream application.	[235,236]
autoMACS/MACS (Magnetic Activated Cell Sorting System)	Mitenyi Biotec, Bergisch Gladbach, Germany	Utilizes an immunomagnetic column to capture cells with diverse antigens (EpCAM, pan-CK, HER2/neu, or CD45), which are retained within the column, and then eluted. Positive selection can be performed by direct or indirect magnetic labeling. Viable cells are available for subsequent analysis following enrichment.	[237,238]
MagSweeper	Stephanie Jeffrey and Ronald W. Davis, Stanford University, Stanford, CA	Automated immunomagnetic device that efficiently capture live CTCs (approximately 0–10 CTCs per 7.5cc tube) from blood, with 100% purity and 60% capture efficiency, while removing contaminating blood cells, and for several downstream analyses. Isolated cells can be extracted individually based on their physical characteristics to deplete any cells nonspecifically bound to beads.	[156,239]
Laser scanning cytometry Maintrac^®^	Simfo, TZB, Bayreuth, Germany	After erythrocyte lysis, the total cell population is analyzed for the presence of circulating epithelial tumor cells, and colored with propidium iodide to differentiate between living and dead cells. Using a fluorescence microscope, the EpCAM-positive cells are automatically identified and counted. A follow-up of serial values allows an assessment of disease activity. Traceable single cell detection within one million cells.	[240,241]
RARE (RosetteSep-Applied Imaging Rare Event)	StemCell Technologies, Vancouver	CTC-negative depletion by targeting CD45 for removal of haematopoietic cells, thus crosslinking cells to multiple red blood cells and generating immunorosettes. Due to higher density of these clusters, they can effectively be separated from CTCs, which are easily collected at the interface between plasma and density gradient medium.	[242]
RoboSep/EasySep™	StemCell Technologies, Vancouver	Column-free immunomagnetic system for isolation of cells for downstream applications. Cells of interest are targeted with antibody complexes and immunomagnetic particles for negative or positive selection and captured. Unlabeled cells are poured off. It is adaptable to custom CTC antibody surface antigens.	
**Functional Assays**			
EPISPOT (EPithelial ImmunoSPOT)	Catherine Alix-Panabieres and Klaus Pantel, Montpellier, France & UKE, Hamburg, Germany	After depletion of CD45 positive cells, by using Rosette plus Ficoll, viable epithelial secreting-cells are identified, at an individual cell level, on a membrane coated with antibodies which detect and measure proteins released from viable CTCs of diverse tumor origins (CK19, MUC1, Cath-D (breast); CK19 (colon); PSA (prostate); TG (thyroid)) by secondary antibodies labelled with fluorochromes. Immunospots are the protein fingerprint left only by viable CTCs.	[243,244]
Vita-Assay^TM^ or Collagen Adhesion Matrix (CAM) technology	Vitatex Inc., Stony Brook, NY	Separation technology is based on preferential adhesion of rare cells in blood of tissue origin to Cell Adhesion Matrix (CAM). CAM enriches viable CTCs in blood, one-million-fold, and identifies invasive CTCs (iCTCs), which express the stem cell marker CD44 and the invasiveness marker seprase, with their capability of ingesting CAM. Once washed, attached cells can be either directly analyzed using microscopy or released from scaffolds for characterization using multi-parameter flow cytometry.	[245,246]
**Dielectrophoresis**			
ApoStream^TM^ System	ApoCell, Houston, TX	It leverages differences in the dielectric properties (polarizability) to isolate cells, using dielectrophoresis (DEP) field-flow fractionation (DFFF). Cancer cells are selectively collected while normal blood cells are carried away by the eluant in a separate port. To use this methodology, an initial enrichment step is required. Recovery rate is over 70% with the viability higher than 97%; the purity obtained is low, and can be improved with additional enrichment stages.	[247,248]
**Density Gradient Centrifugation**			
Ficoll-Paque PLUS	GE Healthcare Life Sciences	Sterile aqueous medium for density gradient centrifugation optimized for purification of mononuclear cells from human peripheral blood, using a simple and rapid centrifugation technique.	[249]
Lymphoprep™	StemCell Technologies, Vancouver	Density gradient medium recommended for the isolation of mononuclear cells from peripheral blood by exploiting differences in cell density. Granulocytes and erythrocytes have a higher density than mononuclear cells and therefore sediment through the Lymphoprep™ layer during centrifugation.	
OncoQuick	Greiner Bio-One, Germany, North Carolina	Liquid separation medium designed to enrich CTC based on their density from up to 30 mL of whole blood. Cells are separated and pass through the barrier, depending on their differentdensities during centrifugation. CTCs, together with lymphocytes, remain above the porous barrier, making them easily accessible for subsequent collection and further processing.	[250,251]
**Microfiltration**			
ScreenCelltechnology (ScreenCell^®^)	ScreenCell Company, Paris	Non-invasive filter-based technology for isolating and sorting circulating rare cells from whole blood, based on cells-size, with a high recovery rate. By providing access to fixed or live CTC and CTC-clusters, it allows phenotypical, genotypical, and functional characterization of cells, with different types of device, depending on the downstream analysis to be performed.	[252,253]
**Microfluidics**			
ClearCell^®^ FX1 System	Biolidics, Singapore	Fully automated and entirely label-free IVD medical device, which relies on a novel patented microfluidic biochip to separate and enrich CTCs from small quantities of whole, unprocessed blood. Intact and viable CTCs are collected in a liquid suspension, can be stained directly on the CTChip for identification or retrieved for further analysis with routine laboratory workflows.	[254,255]
CTC Membrane Microfilter	Richard Cote, Ram Datar, University of Miami, FL	Microdevice for detection and characterization of CTC, based on a stepwise photolithography process that produces filter with controlled-size pores, designed to exploit cell size differences between tumor cells and normal blood cells. Combined with quantum dot-based immunofluorescence detection for CTC characterization.	[256,257]
CTC-Chip	Mehmet Toner and Daniel Haber, Massachusetts General Hospital (MGH) and Harvard Medical School	Microfluidic system that capture large numbers of viable CTCs in a single step from whole blood without pre-processing steps, with high sensitivity, purity, and yield. The first microfluidic device or chip is an array consisting of 78,000 microposts coated with EpCAM antibodies. CTCs, once attached, are visualized and confirmed as CTCs through staining with antibodies. For CTC enumeration, the entire device is imaged using a semi-automated imaging system, while on-chip lysis allowed for DNA and RNA extraction and subsequent molecular analysis.	[7,195,196]
Micropost CTC-Chip	Mehmet Toner and Daniel Haber, Massachusetts General Hospital (MGH) and Harvard Medical School	The technology provided improved yield and purity of captured CTCs isolated. To develop a robust and automated platform capable of high-throughput complex analysis of rare cells, image processing algorithms and scoring criteria were developed to quantify the number of captured CTCs. The digital imaging system with integration of the complete CTC-Chip footprint provided multiplane scanning capacity, a 75% reduction in scanning time, and increased image quality.	[26]
Herringbone CTC-Chip	Mehmet Toner and Daniel Haber, Massachusetts General Hospital (MGH) and Harvard Medical School	This high-throughput microfluidic mixing device constitutes an enhanced platform for CTC isolation where microvortices significantly increase the number of interactions between target CTCs and the antibody-coated chip surface. Cell capture efficiency of the herringbone CTC-Chip is greater, and cells recovered have higher purity compared to micropost CTC-Chip. CTCs can be imaged using clinical histopathological stains, in addition to immunofluorescence-conjugated antibodies.	[7,197]
CTC-iChip	Mehmet Toner and Daniel Haber, Massachusetts General Hospital (MGH) and Harvard Medical School	Integrated and automated microfluidic CTC capture platform, designed with two depletion modes of immunomagnetic sorting to isolate CTCs: a positive one (EpCAM+ cells) and a negative one (CD45+ and CD14+ cells). Nucleated cells (white blood cells and CTCs) are separated, through hydrodynamic size-based sorting, and retained and aligned within the microfluidic channel using inertial focusing; magnetically-labeled cells are separated. Large volumes of whole blood (8 mL per hour), with high throughput (10 million cells per second), are processed at high efficiency. This approach enables cytopathological and molecular characterization of both epithelial and non-epithelial cancers.	[88,219,258,259]
BioFlux	Fluxion Biosciences, South San Francisco, CA	It utilizes the Well Plate Microfluidic™ technology to embed micron-scale fluidic channels on the bottom of a standard well plate, with a uniform and laminar flow profile, ensuring reliability and reproducibility of each experiment. Physiologically-relevant data from cell-based assays are obtained, and data are acquired in brightfield, phase, fluorescence, and confocal high-resolution microscopy.	[260]
Fluidigm Dynamic Arrays for Single-Cell Gene Expression Analysis	Fluidigm Corporation, South San Francisco, CA	Single-cell gene expression technique that, when used with the BioMark™ Real-Time PCR System, allows high-throughput studies on individual cells and is suited to determine single-gene cell expression levels in CTCs.	[261,262]
Parsortix™ cell separation system	ANGLE, Guildford, United Kingdom	Patented step separation microfluidic technology to capture cells based on size and deformability, from 100 µL to 30 mL blood sample, as whole blood flows through steps within the cassette. It requires no sample pre-processing. Examine cells under a microscope in the cassette, or reverse the flow to harvest cells in a test tube for further analysis.	[33,35,263,264]
**Dielectrophoresis**			
DEPArray™ System	Menarini Silicon Biosystems, Bologna	Cell microarray instrument for identification and recovery of individual rare cells. The DEPArray™ cartridge leverages DEPArray™ technology to control the manipulation and collection of cells. The single-use, microfluidic cartridge contains a control circuitry for addressing each individual dielectrophoretic (DEP) cage (cage size can be set to accommodate a single cell). The DEPArray™ analysis platform utilizes high quality, image-based selection to allow identification and sorting of individual, intact cells of interest by morphological parameters on the cartridge. The automated system uses a five-channel fluorescent microscope and camera to capture images and identify cells.	[21,265,266,267,268]
**Other Technologies**			
FASTcell™ Technology	SRI Biosciences, Silicon Valley	Fluorescence cytometry based on Fiberoptic Array Scanning Technology (FAST cytometer), combined with an automated digital microscopy imaging system. Immunofluorescently labelled CTCs, CK positive, are detected on a glass slide using laser-printing optics, which can scan 300,000 cells per second. Rare cells detected by laser scanning up to almost 1,000 times faster than digital microscopy.	[269,270]
Epic Sciences CTC platform	Epic Sciences, Inc.	CTC detection and enumeration in peripheral blood through protein biomarker analysis. Upon blood cell lysis, nucleated cells are deposited on up to 12 microscope slides, immunofluorescently stained, and scanned. A digital algorithm, which includes protein expression and morphology, differentiates CTCs from surrounding white blood cells. CTCs are confirmed and classified as traditional CTCs, CTC clusters, CK(−) CTCs, and apoptotic CTCs. Further, genomic analyses (DNA FISH or next-generation sequencing) are performed if required.	[271,272]
ISET^®^ technology	Rarecells Diagnostics	Unbiased label-free CTC enrichment of all types of CTC by size via vacuum filtration, with high sensitivity (96%) and specificity (98%), maintaining cell morphological and structural integrity. Intact CTC and CTM can be isolated in a fixed or live form and be further exposed to in vitro diagnostic analyses: cytopathological staining, ICC, IF, and molecular analyses (bulk and single-cell).	[273,274]

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
