# Peer review of "Heterogeneity in Circulating Tumor Cells: The Relevance of the Stem-Cell Subset"

_cancers, 2019, doi:10.3390/cancers11040483_

Round 1
Reviewer 1 Report
It is a review paper with 10 authors, I would have doubts regarding the significance of the contribution of all of them. GDL,APDA,PG contributed to the manuscript in the „visualisation” category. There are no images in the manuscript. What was then the input of GDL,APDA,PG? Is their authorship justified? One person wrote the original draft and 7 others (CA,FC,LM,FB,FC,WJJC,SV ) edited and reviewed the manuscript. Was the contribution of these seven people significant enough to be listed as authors?
Maybe listing some of the authors in the Acknowledgements would be enough?
Overall, the review encompasses a large collection of literature on the topic of stem cell markers and EMT markers in circulating tumour cells, but it is lacking a strict structure. Similar information is scattered through different paragraphs, what distracts the flow of thought and blurs the most important messages. Information regarding clinical significance is described in the section “Clinical relevance of heterogeneity in CTCs “, but also in “Methods for CCSC detection”. It is difficult to find a logical flow of information.
What is the main focus of the manuscript? Is it CTCs (as Section 3 “Clinical relevance of heterogeneity in CTCs “ would indicate) or CCSC (as Section 4 “Methods for CCSC detection” would imply)?
Specific comments to the manuscript I have listed below.
1. Line 61-63 “A lot of evidence document that CTCs represent a heterogonous pool of tumour cells [8], with highly variable survival rates, from less than a few hours to decades [21,22].”
CTCs can be detected decades after diagnosis and removal of a primary tumour, but it does not mean that they circulate for decades (and therefore this is their survival time). They might be released to the bloodstream from secondary small metastatic foci. Also, hours to decades is a measure of survival time rather than survival rate. Please correct.
2. Line nr 78-79. “Mesenchymal transcripts were expressed in tumour cells of CTC-clusters in human breast 79 tumour, conferring both migratory and stem-like properties [14,32].” Please check again in in reference nr 14 CTCs clusters were really evaluated.
3. Please add also study by Yu at all to the section “2.1. Partial mesenchymal transition in CTCs” Yu, M.; Bardia, A.; Wittner, B.S.; Stott, S.L.; Smas, M.E.; Ting, D.T.; Isakoff, S.J.; Ciciliano, J.C.; Wells, M.N.; 555 Shah, A.M.; et al. Circulating breast tumor cells exhibit dynamic changes in epithelial and mesenchymal composition. Science 2013, 339, 580–584.
4. Line 169-170. The link between lin-7 and EMT has to be described.
5. Please correct the sentence “Consistent, CTCs expressing detectable levels of EPCAM and cytokeratins next to CD44, CD47, and MET, generated, in immunodeficient mice, metastases, with an EPCAMlowMEThighCD47highCD44high phenotype, characteristic of metastasis-initiator cells [33].”
6. Line 188-189 “The CSC phenotypes need to be confirmed in early stage cancer patients and prospectively related to the occurrence of overt metastases. “
There are already publications showing expression of CSC markers in CTCs of early breast cancer patients (PMID: 25182808, PMID: 29660692).
7. Please explain the meaning of the sentence – does it mean that HER2 triggers cell division of CSC? “Modulation of HER2 signaling can induce duplication of stem cells”
8. Line 214-218 “At least one or more of the EMT markers TWIST, Akt2, and PI3Kα, and ALDH1 were detectable in a great proportion of CTCs, thus identifying the highly tumorigenic subset of EMT- associated breast CSCs [45]. This subset of CTCs is indicative of therapy-resistance and inferior prognosis in metastatic tumour patients, with clinical relevance [45]. Of note, ALDH1 and EMT markers were measured even before detection of CTCs in circulation [45].”
In the last sentence “ALDH1 and EMT markers were measured even before detection of CTCs in circulation” – where exactly were ALDH1 and EMT markers found if patients were negative for CTCs?
9. Line 239-240 “Most breast tumour patients have CK+/CD44+/CD24-/low CTCs [58], but CTCs expressing CD44, MET and CD47 might also represent CSCs [33].”
This sentence is misleading as CTCs phenotype CK+/CD44+/CD24-/low was common only in the cited study, it is not known if such phenotype occurs frequently in other cases. If most of CK+ CTCs have this stem cell phenotype ( CD44+/CD24-/low) then EMT would not be necessary to induce stemness.
10. It would be good if Table 1 grouped studies according to cancer type.
11. Line 307-308 “upon targeted therapy two distinct CTC subgroups were induced, which differ in both AR genotype and clinical phenotype [156].” What is “clinical phenotype”? Please explain.
12. Chapter “3. Clinical relevance of heterogeneity in CTCs” contains very widespread set of information regarding CTCs. Topics regarding thereapy-relavant markers (ER/HER2), EMT markers, CSC markers, mutations in proto-oncogenes are described in the context of clinical significance. The focus of the paper, which is meant to be CSC is therefore diffused.
13. Line 369-372 “In metastatic breast cancer patients, expression in CTCs of at least one EMT marker and ALDH1 was detected in 62% and 44%, respectively, of patients non responders to therapy, suggesting their potential use as predictive biomarkers [45]”. Please discuss what therapeutic regimen was used.
14. Line 435 “Overall, only a limited success in CCSC detection has been reported [213,225].” To which methods this statement refers?
15. Line 439 “Five different RANs were designed that identify different CTC subtypes and CCSC's stemness, by targeting CD133.” Please explain what RANs are.
16. Line 451 ”Plastin 3 (PLS3) overexpression in CTCs has been reported to induce EMT+ CTCs in CRC”. Plastin 3 is a marker of EMT undergoing cells, but it does not induce EMT.
17. Lines 467-469 “Recently we described a method based on a qRT-PCR assay, with a great potential for the study of CTC heterogeneity and for the identification of novel CTC markers, irrespective of currently used markers and without pre-selection or enrichment, by using patient-derived xenografts (PDXs) models [230].” It should be more clearly stated that the method does not apply to the analysis of patient’s blood samples as it is basing on the difference in gene sequences between mouse and human.
Author Response
Special thanks to Reviewer #1 for his/her thorough reading, and thoughtful critiques, not only of the details, but also for the whole structure of our manuscript.
Please, find in attachment all comments, annotations and revision to the manuscript according to your notes, as detailed for each note.

Reviewer 2 Report
This is a well designed study aimed to addresses EMT and cancer stem cells related CTC in heterogeneity. This is also well organized and well written manuscript which will be ready to publish in the journal.
Author Response
We are grateful to Reviewer #2 for his/her positive comments of our manuscript, both for the novelty of the subject we discussed, addressing mesenchymal and stem-like CTCs sub-populations, and the structure of the text.
This is a well designed study aimed to addresses EMT and cancer stem cells related CTC in heterogeneity. This is also well organized and well written manuscript which will be ready to publish in the journal.
Furthermore, we now revised the manuscript as suggested from the other reviewers, thus emphasizing specific aspects of stem-cell like CTC, from a clinical point of view and relative to methodologies for their inclusive characterization.

Reviewer 3 Report
This review descrive in a very exhaustive way the complex scenario of mesenchymal and stemness markers involved in CTC detection, and indeed Table 1 represents a useful summary (even for experts in the field) of the most relevant markers, associated with distinct tumor types, with a detailed description of their function and clinical relevance.
What has not been sufficiently detailed is the number of approaches currently available to enrich and isolate CTCs, mainly for a molecular characterization (that represents the most promising tool derived from CTCs for precision medicine). I think that a Table, summarizing such approaches (and indeed the possibility to detect even double positive and double negative cell subpopulations, whose role on tumor biology and metastatization process is becoming relevant, in these last days) could be of markedly valuable for the investigators, particularly for those not familiar with the complex scenario of liquid biopsies and CTCs.
Author Response
Many thanks to Reviewer #3 for his/her comments about our review.
This review describes in a very exhaustive way the complex scenario of mesenchymal and stemness markers involved in CTC detection, and indeed Table 1 represents a useful summary (even for experts in the field) of the most relevant markers, associated with distinct tumor types, with a detailed description of their function and clinical relevance.
What has not been sufficiently detailed is the number of approaches currently available to enrich and isolate CTCs, mainly for a molecular characterization (that represents the most promising tool derived from CTCs for precision medicine). I think that a Table, summarizing such approaches (and indeed the possibility to detect even double positive and double negative cell sub-populations, whose role on tumor biology and metastatization process is becoming relevant, in these last days) could be of markedly valuable for the investigators, particularly for those not familiar with the complex scenario of liquid biopsies and CTCs.
As suggested by this Reviewer, we included a Table (Table 2) listing the different approaches described in literature for enrichment and isolation of CTCs, with particular focus on those technologies allowing a molecular characterization of CTCs, for clinical applications.

Reviewer 4 Report
The manuscript reviews the current literature in emt/stemness in ctcs. It is a well written manuscript which needs a few items to be addressed.
In section 2, the authors discuss the recent CTC cluster data in breast cancer. Can the authors comment on other tumour types in this space? Head and neck cancer - Kulasinghe et al., Scientific reports 2018;Cancers 2019; Aceto et al., glioblastoma ctc clusters - Aceto et al., British journal of cancer 2018)? Additionally, there is another recent manuscript by the Aceto group which describes that neutrophils escort ctc clusters - this should be included.
Lines 182 - the authors describe epcam + CTCs - this should be highlighted as only epcam positive ctcs have demonstrated to be clinically relevant despite other technologies coming to the fore. Are the emt-transitioned CTCs clinically relevant? These points should be discussed.
Author Response
Thanks to the Reviewer #4 for his/her positive comments on our manuscript.
Please, find in attachment all comments, annotations and revision to the manuscript according to your notes, as detailed for each note.

Round 2
Reviewer 1 Report
Though the authors tried to improve the manuscript, added new data and responded to the comments, I still think that overall the manuscript lacks a clear structure and a sharp focus.